# Effect of Exercise-Based Cardiac Rehabilitation on Left Ventricular Function in Asian Patients with Acute Myocardial Infarction after Percutaneous Coronary Intervention: A Meta-Analysis of Randomized Controlled Trials

**DOI:** 10.3390/healthcare9060774

**Published:** 2021-06-21

**Authors:** Yanjiao Wang, Ching-Wen Chien, Ying Xu, Tao-Hsin Tung

**Affiliations:** 1Institute for Hospital Management, Tsing Hua University, Shenzhen Campus, Shenzhen 518055, China; yj-wang20@mails.tsinghua.edu.cn (Y.W.); ihhca@sz.tsinghua.edu.cn (C.-W.C.); xuy20@mails.tsinghua.edu.cn (Y.X.); 2Evidence-based Medicine Center, Taizhou Hospital of Zhejiang Province Affiliated to Wenzhou Medical University, Linhai 317000, China

**Keywords:** acute myocardial infarction percutaneous coronary intervention, exercise, rehabilitation, left ventricular function

## Abstract

(1) Background: The effects of exercise-based cardiac rehabilitation (CR) on left ventricular function in patients with acute myocardial infarction (AMI) after percutaneous coronary intervention (PCI) are important but poorly understood. (2) Purpose: To evaluate the effects of an exercise-based CR program (exercise training alone or combined with psychosocial or educational interventions) compared with usual care on left ventricular function in patients with AMI receiving PCI. (3) Data sources, study selection and data extraction: We searched PubMed, WEB OF SCIENCE, EMBASE, EBSCO, PsycINFO, LILACS and Cochrane Central Register of Controlled Trials databases (CENTRAL) up to 12th June 2021. Article selected were randomized controlled trials and published as a full-text article. Meta-analysis was conducted with the use of the software Review manager 5.4. (4) Data synthesis: Eight trials were included in the meta-analysis, of which three trials were rated as high risk of bias. A significant improvement was seen in the exercise-based CR group compared with the control group regarding left ventricular ejection fraction (LVEF) (std. mean difference = 1.33; 95% CI:0.43 to 2.23; *p =* 0.004), left ventricular end-diastolic dimension (LVEDD) (std. mean difference = −3.05; 95% CI: −6.00 to −0.09; *p* = 0.04) and left ventricular end-systolic volume (LVESV) (std. mean difference = −0.40; 95% CI: −0.80 to −0.01; *p* = 0.04). Although exercise-based CR had no statistical effect in decreasing left ventricular end-systolic dimension (LVESD) and left ventricular end-diastolic volume (LVEDV), it showed a favorable trend in relation to both. (5) Conclusions: Exercise-based CR has beneficial effects on LV function and remodeling in AMI patients treated by PCI.

## 1. Introduction

Acute myocardial infarction (AMI) is a common cardiac emergency caused by myocardial necrosis resulting from hypoxia and ischemia [1]. It retains the potential for substantial morbidity and mortality worldwide [2], which causes more than 216,000 deaths in the USA, and more than one third of deaths in developed countries annually [3,4]. Percutaneous coronary intervention (PCI) is an effective treatment for AMI and has a favorable early and long-term prognosis [5]. It can rapidly restore myocardial reperfusion, quickly alleviate myocardial hypoxia/ischemia, and reduces AMI patients’ mortality [6]. Although PCI can quickly relieve symptoms, many patients still suffer from myocardial damage, poor mental state and decreasing motor ability [7,8].

Exercise-based cardiac rehabilitation (CR) has significant benefits for survival, quality of life (QOL) and psychological health in AMI patients receiving PCI that have been widely proven [9]. However, the impact of exercise-based CR on left ventricular (LV) function in AMI after PCI is uncertain. Some prior trials showed that exercise-based CR led to decreases in left ventricular end-diastolic dimension (LVEDD), left ventricular end-systolic dimension (LVESD), left ventricular end-diastolic volume (LVEDV) and left ventricular end-systolic volume (LVESV) and increases in left ventricular ejection fraction (LVEF) [10,11,12]. In other trials, exercise-based CR does not alter LVEF, LVEDD, LVESD, LVEDV or LVESV [13,14,15]. We therefore conducted a systematic review and meta-analysis to evaluate the overall effectiveness of exercise-based CR on LV function in AMI patients treated by PCI. The hypothesis of this study is that exercise-based CR compared with usual care has beneficial effects on LV function in patients with AMI after PCI.

## 2. Materials and Methods

### 2.1. Searching for Literature

A search was undertaken of PubMed, WEB OF SCIENCE, EMBASE, EBSCO, PsycINFO, LILACS and CENTRAL for relevant studies with no language limitations on 12 June 2021. Searches included a mix of MeSH and free-text terms related to the key concepts of acute myocardial infarction, percutaneous coronary intervention, cardiac rehabilitation and left ventricular function (Table 1).

### 2.2. Study Selection

Randomized controlled trials (RCTs) that examined the effectiveness of exercise-based CR on LV function in AMI patients treated by PCI were included. Two investigators scanned the titles and abstracts of all potential studies and identified suitable studies that met our selection criteria independently. Disagreement was resolved through consensus from a third investigator.

### 2.3. Data Extraction and Risk of Bias Assessment

Two authors extracted relevant outcome data from the included studies independently and any disagreement was resolved by consensus in discussion with the third author. The primary outcome was LV function. The measure of effect used was the left ventricular ejection fraction (LVEF), left ventricular end-diastolic dimension (LVEDD), left ventricular end-systolic dimension (LVESD), left ventricular end-diastolic volume (LVEDV) and left ventricular end-systolic volume (LVESV). We assessed the quality and the risk of bias of included RCTs according to Cochrane Collaboration’s tool which included selection bias, performance bias, detection bias, attrition bias, reporting bias and other bias. In the same way, any disagreement was adjudicated by the third author.

### 2.4. Statistical Analysis

Statistical analysis was conducted using the Review Manager 5.4 (The Nordic Cochrane Centre, The Cochrane Collaboration, Copenhagen, Denmark, 2020). The outcomes of intervention effect were evaluated by echocardiograph at baseline and after intervention. The mean differences and 95% confidence intervals (CI) were used to represent the intervention effect. Statistical heterogeneity was quantified by the chi-square test and the I^2^ statistic. Given *p* ≤ 0.10, I^2^ ≥ 50%, we adopted the random-effects model would be adopted; otherwise, a fixed-effects model would be applied. The results were presented as the standardized mean difference (SMD) with 95% CI and *p* value, *p* < 0.05 was considered to be statistically significant. Publication bias was assessed by funnel plot.

## 3. Results

### 3.1. Characteristics of Included Studies

We initially retrieved 2816 articles through the electronic database searches, with 1801 remaining after removing 1015 duplicates. Based on inclusion criteria, 47 full-text articles were evaluated for eligibility. The meta-analysis ultimately included eight [14,16,17,18,19,20,21,22] of them (Figure 1). A total of 582 participants were enrolled in the included studies. The characteristics of these eight studies are illustrated in Table 2. Six studies were conducted in China [14,16,17,18,19,22], and one each in Japan [20] and Iran [21]. All the included studies reported the LVEF, three for LVEDD [17,19,21], two for LVESD [19,21] and two for LVEDV and LVESV [14,21].

The assessment of the risk of bias for all included studies was summarized and shown as Figure 2. The overall risk of bias was low or unclear. All studies presented balance in baseline characteristics. Almost all the studies reported that the study was ‘randomized’ while three trials did not provide the details of the generation of the random sequence [16,20,22]. Only two studies reported appropriate concealment of allocation, other studies [16,17,19,20,21,22] were rated unclear due to lack of sufficient details. In the aspect of blinding, it is impractical to blind participants and program personnel on the basis of exercise-based CR. Four articles [14,18,20,21] reported adequate description of the blinding of the outcome assessment. For attrition bias, only one study [18] was judged as “high risk of bias” due to 14.6% rate of loss of follow-up. The risk for reporting bias was low for all included studies. Two articles [19,22] were rated “uncertain risk of bias” for other bias because of a lack of sufficient information.

### 3.2. Exercise-Based CR and LVEF

All the included studies with a total of 582 participants provided data on LVEF; we found that the benefit in the experimental group was greater than the control group (std. mean difference = 1.33; 95% CI:0.43 to 2.23; *p* = 0.004) by random-effects model (*p* < 0.0001, I-squared = 95%) (Figure 3).

### 3.3. Exercise-Based CR and LVEDD

Three of the studies [17,19,21] focused on the effect of exercise-based CR on LVEDD of patients with AMI after PCI. The result showed that exercise-based CR had a significant effect in reducing LVEDD compared with the control group (std. mean difference = −3.05; 95% CI: −6.00 to −0.09; *p* = 0.04). Based on the high level of heterogeneity (*p* < 0.0001, I-squared = 98%) we used a random-effects model to analyze data (Figure 4).

### 3.4. Exercise-Based CR and LVESV

Additionally, two studies [14,21] showed that exercise-based CR has a positive effect in reducing LVESV. Figure 5 showed that there was a more significant decrease in LVESV in the experimental group than in the control group (std. mean difference = −0.40; 95% CI: −0.80 to −0.01; *p* = 0.04). The heterogeneity has no significant difference (*p* = 0.047, I-squared = 0%).

### 3.5. Exercise-Based CR and LVESD, LVEDV

LVESD [19,21] and LVEDV [14,21] each had two studies to present data. Meta-analysis showed no statistically significant decrease in LVESD (std. mean difference = −0.29; 95% CI: −0.67 to 0.09; *p* = 0.144) and no statistical effect in decreasing LVEDV (std. mean difference = −0.10; 95% CI: −0.49 to −0.29; *p* = 0.60), but showed favorable trends. Statistical heterogeneity across these studies was low (I-squared = 0%) (Figure 6 and Figure 7).

## 4. Discussion

### 4.1. Clinical Implications

To the best of our knowledge, this is the first meta-analysis to evaluate the effect of exercise-based CR on LV function in patients with AMI who received PCI. In this meta-analysis we assessed the evidence from RCTs that compared outcomes with the exercise-based CR and the control. We found that the exercise-based CR did significantly improve the LV function as indicated by the significant increase in LVEF and decrease in LVEDD and LVESV as compared with the control. Although the change in LVEDV and LVESD between the exercise-based CR and the control was not statistically significant, a favorable trend was shown in the participants of the exercise-based CR group.

After decades of research and development, exercise-based CR has been gradually applied in cardiac patients and its benefit has been widely proven by clinical research evidence. A meta-analysis of sixty-three studies [23] showed that exercise-based CR can effectively reduce cardiovascular mortality and the overall risk of hospital admissions in patients with coronary heart disease. Dugmore and his colleagues [24] have indicated that exercise-based CR can elicit improvements in QOL and psychological wellbeing in post myocardial infarction patients. Several studies have also shown that exercise-based CR is beneficial for preventing LV remodeling [25,26,27,28] and improving LV function [29,30] after myocardial infarction. Indeed, our results further supported that exercise-based CR can have positive impact on LV function in AMI patients treated by PCI.

Cardiac rehabilitation not only emphasizes exercise training, but is also a comprehensive secondary prevention program consisting of structured exercise, dietary education, psychological counselling and risk factor management [31,32]. The settings of CR delivery that include home-based and center (or hospital)-based current evidence support that those two have equal effects on improving clinical outcomes [33,34,35]. Although CR has been demonstrated to have beneficial effects, its development as a treatment has been poor, with <25% participation [36,37,38] and a more than 50% drop-out rate by 1 year [39]. Not participating or not sticking to a CR program is associated with patient factors, healthcare professional factors and accessible factors [40,41]. Promoting participation in CR still has a long way to go; referral by a cardiologist has a positive effect on improving participation [42,43].

### 4.2. Methodological Considerations

The strength of our study is that we analyzed the effectiveness of exercise-based CR on LV function in patients with AMI who received PCI. A prior meta-analysis only analyzed the effect of exercise-based CR on LV function in patients treated by PCI [44]. Zhang et al. conducted a meta-analysis to assess the effect of exercise-based CR on LV function in patients after myocardial infarction [45]. It means that compared to them, our research may show a more comprehensive result.

Several potential limitations of this meta-analysis deserve mention. Firstly, the onset and duration of CR varied among the included studies, the initiation of CR varied from immediately post-PCI to one month post-PCI, with the duration lasting from 7 days to 6 months. Several meta-analyses have shown that the onset and duration of CR may influence the effectiveness of the treatment [28,45]; thus, future trials are needed to evaluate the specific effects of these factors. Secondly, the studies were included in the LVEF and LVEDD forest plots showing high heterogeneity. In this study, we used the random-effect model when I-squared statistics were 95% and 98% of LVEF and LVEDD more than 50%, respectively. For the high degree of heterogeneity, we conduct a sensitivity analysis to evaluate the reliability of the results. The result indicates that the problem may be caused by the short duration of CR [17], small sample numbers [20] or included few studies (only three studies included in the LVEDD). Thirdly, in the case of LVESD, LVEDV and LVESV there were only two studies available; results from such a small sample size of studies are more subject to chance [46] and this may cause the funnel plot to be difficult to interpret, making it more difficult to detect publication bias [47]. Therefore, the strength of the conclusions may be questioned; therefore, we hope there will be more trials to assess the effect of CR in the future. Fourthly, all of the included trials are from Asian countries, a fact which limits the scope of application and power of the findings. However, some research results from other continents were consistent with our findings. Volodina and his colleagues [12] from Russia showed that cardiac rehabilitation has beneficial effects on LV function in NSTEACS (including AMI) patients treated by PCI. In addition, by virtue of the limited number of articles included, and the lack of sufficient data, it was difficult to conduct subgroup analyses. Finally, in most of the RCTs included, the allocation concealment was poorly reported, which led to an increase in the risk of selection bias in the results.

## 5. Conclusions

In summary, the current evidence showed that exercise-based CR has favorable effects on LV function and remodeling in AMI patients after PCI as indicated by the significant increase in LVEF and decrease in LVEDD and LVESV. Enlarging the sample size and evaluating the specific effects of onset and duration of CR will be very important in future study.

## Figures and Tables

**Figure 1 healthcare-09-00774-f001:**
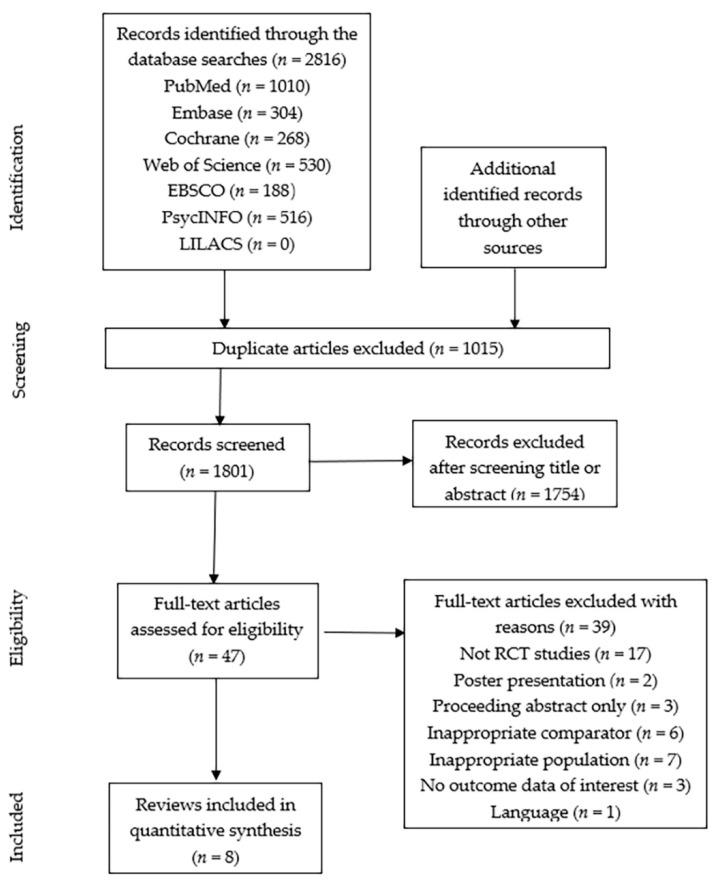
PRISMA flow diagram.

**Figure 2 healthcare-09-00774-f002:**
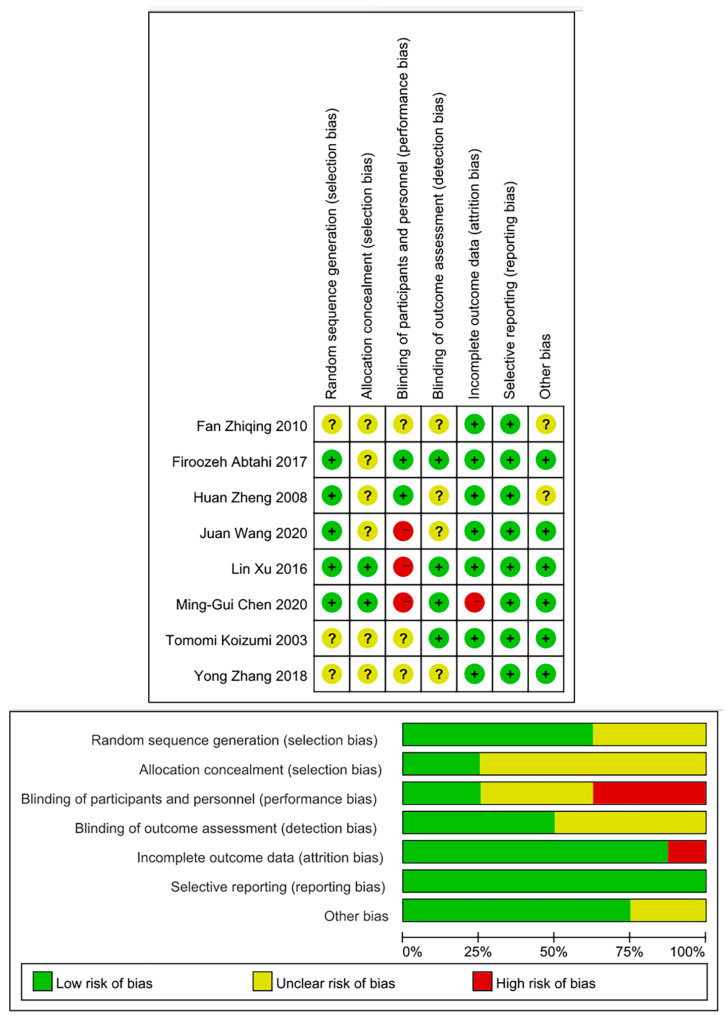
Risk of bias summary.

**Figure 3 healthcare-09-00774-f003:**
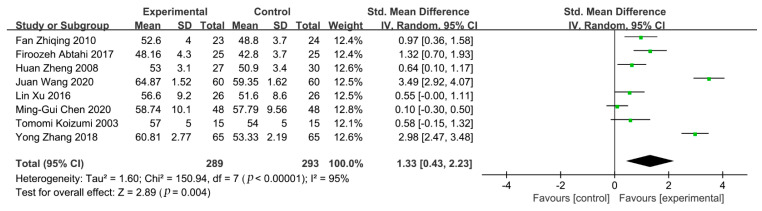
Meta-analysis result of LVEF.

**Figure 4 healthcare-09-00774-f004:**
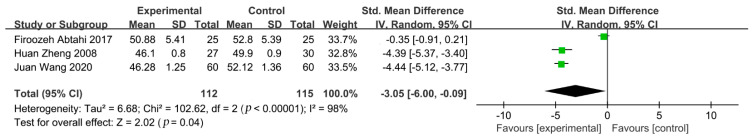
Meta-analysis result of LVEDD.

**Figure 5 healthcare-09-00774-f005:**
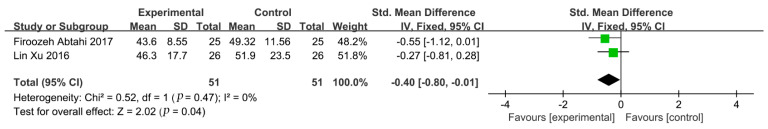
Meta-analysis result of LVESV.

**Figure 6 healthcare-09-00774-f006:**
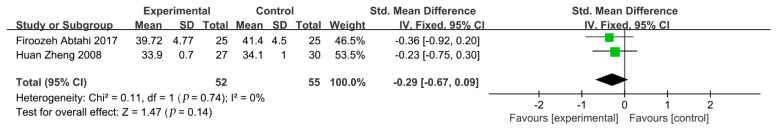
Meta-analysis result of LVESD.

**Figure 7 healthcare-09-00774-f007:**
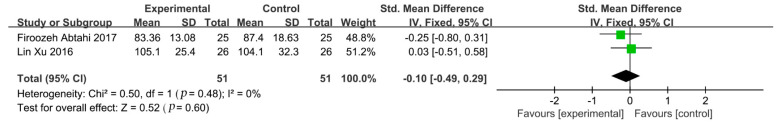
Meta-analysis result of LVEDV.

**Table 1 healthcare-09-00774-t001:** Search strategy until 12 June 2021.

#1	acute myocardial infarction
#2	AMI
#3	acute Heart attacks
#4	acute coronary syndromes
#5	ACS
#6	#1 OR #2 OR #3 OR #4 OR #5
#7	percutaneous coronary intervention
#8	PCI
#9	revascularize
#10	#7 OR #8 OR #9
#11	cardiac rehabilitation programs
#12	CRP
#13	cardiac rehabilitation
#14	CR
#15	physical training
#16	exercise training
#17	exercise therapy
#18	exercise
#19	kinesiotherapy
#20	rehabilitation
#21	mobilization
#22	#11 OR #12 OR #13 OR #14 OR #15 #16 OR #17 OR #18 OR #19 OR #20 OR #21
#23	left ventricular function
#24	ventricular remodeling
#25	myocardial function
#26	diastolic function
#27	ventricular volumes
#28	Left ventricular ejection fraction
#29	LVEF
#30	EF
#31	left ventricular end-diastolic dimension
#32	left ventricular end-systolic dimension
#33	left ventricular end-systolic volume
#34	left ventricular end-diastolic volume
#35	LVEDD
#36	LVESD
#37	LVEDV
#38	LVESV
#39	#23 OR #24 OR #25 #26 OR #27 OR #28 OR #29 OR #30 OR #31 OR #32 OR #33 OR #34 OR #35 #36 OR #37 OR #38
#40	#6 AND #10 AND #22 AND 39

**Table 2 healthcare-09-00774-t002:** Characteristics of included studies.

Author, Publication, Year	Country	Study Period	Assigned Group	Participants Characteristics	Exercise Intervention			Major Findings
					Type of Excises	Frequency/Session Duration/Intensity	Onset/Total Duration	
Yong Zhang et al., 2018 [16]	China	2010–2012	Exp: CR based on routine therapy	*n* = 65, age 70.3 ± 10.7 years, 90.8% males	walk and other aerobic exercise	Phase II: 2–3 times per week/15–30 min (+10 min warm-up and 10-min cool-down)/HR < 130 bpm or resting HR plus 30 bpm, 250–300 kcal/time; phase III: 3–5times per week/30–45 min (+10min warm-up and 10-min cool-down)/60–75% HRmax, 300–400 kcal per time.	Phase II: the second week after discharge/6–8 weeksphase III: the 3rd month to the 6th month/4 months	Exp vs. Con: ⬆LVEF, *p* < 0.01
			Con: usual care and conventional drug therapy	*n* = 65, age 69.8 ± 10.4 years, 83.1% males	usual care and conventional drug therapy		NA	
Juan Wang, 2020 [17]	China	2017–2018	Exp: CR care	*n* = 60, age 60.28 ± 2.82 years, 51.67% males	24h after surgery: passive movement and deep breathing exercise; 1 day after surgery: sit at bedside; 2–7 days after surgery: walk in the ward	24h after surgery: not specified/not specified/not specified; 1 day after surgery: 3times per day/<30 min/not specified; 2–7 days after surgery:3 times per day/walk 40–300 m/not specified.	Immediately post-PCI/7 days	Exp vs. Con: ⬆LVEF, *p* < 0.001; ⬇LVEDD, *p* < 0.001.
			Con: routine care	*n* = 60, age 59.36 ± 3.27 years, 58.33% males	basic clinical care, monitoring condition, performing routine drug therapy according to medical orders and gradually increasing physical activity after 3 days of bed rest.		NA	
Ming-Gui Chen et al., 2020 [18]	China	2016–2017	Exp: received 24 weeks of BST training	*n* = 48, age 59.98 ± 10.86 years, 67.4% males	Baduanjin exercise or regular aerobic exercise	during hospitalization: 2 times per day/30 min/not specified; Discharge: 5 times per week/30 min/not specified.	second day post-PCI/during hospitalization: 3 days Discharge: lasting up to 24-weeks	⬌LVEF
			Con: no training	*n* = 48, age 61.49 ± 11.54 years, 76.9% males	requested to maintain original habit of lifestyle.		NA	24-week vs. Baseline: ⬇LVEF, *p* = 0.020
Huan Zheng et al., 2008 [19]	China	unclear	Exp: followed a 6-month exercise program	*n* = 27, sex and age are unknown	exercise performed on a bicycle ergometer	3 times per week/30 min (+15min warm-up and 15-min cool-down)/not specified.	3–7 days post-primary PCI/6 months	Exp vs. Con: ⬆LVEF, *p* = 0.003; ⬇LVEDD, *p* = 0.018; ⬌LVESD
			Con: received routine recommendations	*n* = 30, sex and age are unknown	received routine pharmacological therapy and lifestyle education		NA	
Lin Xu et al., 2016 [14]	China	2014–2015	Exp: early, home-based CR program	*n* = 26, age 55.8 ± 9.7 years, 84.6% males	inpatient phase: casual limb movements in bed and simple walk training outpatient phase: aerobic exercise (i.e., walking or jogging, gymnastics)	inpatient phase:2–4 times per day/10–20 min/2 to 4METs, 60% HRmax; outpatient phase: 3 times per day/15–30 min (+5 min warm-up and 5 min cool-down)/60%HRmax.	Immediately post- PCI/inpatient phase: 7–10 days; outpatient phase: 4 weeks	Exp vs. Con: ⬆LVEF, *p* = 0.008; ⬌LVESV ⬌LVEDV
			Con: usual care	*n* = 26, age 55.5 ± 8.9 years, 84.6% males	usual care program including physical activity		NA	
Tomomi Koizumi et al., 2003 [20]	Japan	1998–1999	Exp: exercise training program	*n* = 15, age 54 ± 12 years, 92.86% males	walking	every day/>30 min/not specified	One month post- PCI/3month	⬌LVEF
			Con: educational support	*n =* 15, age 59 ± 9 years, 86.67%males	educational support, avoid strenuous physical activity			
Firoozeh Abtahi et al., 2017 [21]	Iran	2015–2016	Exp: CRP	*n =* 25, age 53.76 ± 6.96 years, 56% males	aerobic exercise	3 times per week/40 min (+10 min warm-up and 10-min cool-down)/40–60%HRR	1 to 2 weeks after AMI/8weeks	follow-up vs. Baseline: ⬆LVEF, *p* < 0.001; ⬇LVEDD, *p* = 0.047; ⬇LVESD, *p* < 0.001; ⬇LVESV, *p* < 0.001; ⬌LVEDV
			Con: instructed on risk factor management	*n =* 25, age 53.6 ± 6.98 years, 60% males	instructed on risk factor management			follow-up vs. Baseline: ⬌LVEF ⬌LVEDD⬌LVESD ⬌LVEDV ⬌LVESV
Fan Zhiqing et al., 2010 [22]	China	2008–2009	Exp: rehabilitation exercise	*n =* 47(Exp = 23, Con = 24), age 62.0 ± 5.6 years, 84% males	aerobic exercise	1–2 times per day,4–5 days per week/<30 min/60–80% HRmax	2–4 weeks after AMI/6 months	follow-up vs. Baseline: ⬆LVEF, *p* < 0.001
			Con: usual care		no exercise prescription and no exercise rehabilitation guidance			follow-up vs. Baseline:⬌LVEF

⬌ non-significant change, ⬆significant increase, ⬇significant decrease, Exp experimental, Con control, CR cardiac rehabilitation, CRP cardiac rehabilitation program, HR heart rate, HRmax maximum heart rate, METs metabolic equivalents, HRR heart rate reserve.

## Data Availability

All data underlying the findings are within the paper.

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
