# Peer review of "Effect of Exercise-Based Cardiac Rehabilitation on Left Ventricular Function in Asian Patients with Acute Myocardial Infarction after Percutaneous Coronary Intervention: A Meta-Analysis of Randomized Controlled Trials"

_healthcare, 2021, doi:10.3390/healthcare9060774_

Round 1

Reviewer 1 Report

This manuscript is written well, so I recommend to publish after minor revision.

1 ) The purpose of the present study was to assess the effectiveness of exercise-based CR (exercise training alone or in combination with psychosocial or educational interventions) compared with usual care on left ventricular function in patients with AMI after PCI. However, there is no description of the hypothesis of this study. Please describe in addition. 

2) In figure 2, this is not easy to see for readers, so could you revise this more?

3)  From figure 3 to Figure 7 are also not easy to see. 

4) Are there any other limits to research? 

Author Response

1. The purpose of the present study was to assess the effectiveness of exercise-based CR (exercise training alone or in combination with psychosocial or educational interventions) compared with usual care on left ventricular function in patients with AMI after PCI. However, there is no description of the hypothesis of this study.Please describe in addition.

Response1: Thanks for the reviewer’s useful comments. We apologize for the inadequate descriptions. The hypothesis of this study is that exercise-based CR compared with usual care have beneficial effect on LV function in patients with AMI after PCI. We have added the hypothesis into our study.

2. In figure 2, this is not easy to see for readers, so could you revise this more?

Response 2: Thanks for the reviewer’s useful comments. We have revised this figure for easy to see. Please see the attachment.

3. From figure 3 to Figure 7 are also not easy to see. 

Response 3: Thanks for the reviewer’s useful comments. We have revised these figures for easy to see. Please see the attachment.

4. Are there any other limits to research?

Response 4: Thanks for the reviewer’s useful comments. We apologize for the inadequate descriptions. The limitations have been corrected. Please see line 243-251.

Reviewer 2 Report

Methods

Why weren't other databases, specific to some continents, not included? For example, Lilacs is interesting because it contemplates Latin American literature. 

Discussion

Line 216 - What is an adequate sample size? How did the small sample sizes of the studies influence the analysis? 

Author Response

1. Why weren't other databases, specific to some continents, not included? For example, Lilacs is interesting because it contemplates Latin American literature. 

Response 1: Thanks for the reviewer’s useful comments. Based on your suggestions and some meta-analysis of cardiac rehabilitation, we have added EBSCO, PsycINFO and LILACS databases. The descriptions also have been corrected.

2. Line 216 - What is an adequate sample size? How did the small sample sizes of the studies influence the analysis? 

Response 2: Thanks for the reviewer’s useful comments. We apologize for the inadequate descriptions. The limitations have been corrected. Please see line 239-241.
